

# Different sound exposures causes alterations in stress-related serum indicators, behaviors, and cecal microbiota of green-shell egg-laying chickens under different stocking densities

Shiwen Cao*, Manhong Ye*, Wanhong Wei and Fengping Yang

Department of Animal Behavior, College of Bioscience and Biotechnology, Yangzhou University, Yangzhou, Jiangsu, China
* These authors contributed equally to this work.

Corresponding author
Fengping Yang, fpyang@yzu.edu.cn

## ABSTRACT

Sound and stocking density are two common factors which influence the performance and welfare of layers. Accumulated studies have been conducted on the impacts of the two factors on production performance, while knowledge regarding the impacts of the two factors and their interactions on stress-related serum indicators, behaviors, and cecal bacterial communities in laying hens is still limited. A 3 × 3 factorial design with three sound sources (natural sound (NS), instrumental music (IMS), or mixed road noise (MRS)) and three stocking densities (low density (LD), medium density (MD), and high density (HD)) was used in this 24-day experiment, in which 378 30-week-old Xiandao green-shell layers were randomly distributed into nine treatments with six replicates per treatment. At the 3rd, 12th, and 24th experimental day, we evaluated the serum levels of adrenocorticotropic hormone (ACTH) and corticosterone (CORT) and recorded stress-related animal behaviors. At the end of the experiment, 16S rRNA gene amplicon sequencing of the cecal bacterial communities was performed. Our results confirmed that MRS and HD induced significantly elevated serum ACTH and CORT levels, and were correlated with significantly increased feather pecking behavior. IMS and LD were associated with enhanced preening behavior and reduced feather pecking behavior. LD significantly increased the Firmicutes/Bacteriodetes ratio and IMS significantly enriched the beneficial *Lactobacillus* population. Based on the obtained results we proposed that music exposure and reduced stocking density were helpful in reducing stress and improving cecal bacterial profile, which were beneficial for improving layers' health status and welfare.

## INTRODUCTION

In China, intensive cage rearing is currently the main rearing mode adopted by conventional chicken farms due to its low cost, easy management, and high profitability. According to the stipulation of European Union, the stocking density of laying hens should

not be less than 750 cm$^2$/bird (*Kang et al., 2016*), while the commonly adopted stocking density of layers in China is around 450–500 cm$^2$/bird. In addition, with the upgrading and optimization of farms, the stocking density of some laying hens could be reduced to a higher degree owing to the lack of space. The limited space accessible for the intensively reared layers has brought about series of concerns, including the increased incidence of diseases (*Fang et al., 2021*), elevated stress level and reduced animal welfare (*Kaya et al., 2021*).

In the meantime, chicken farms are usually located not far from highways in order to facilitate the transportation of hens and their products. Therefore, chickens are inevitably under the exposure of two negative environmental factors, high stocking density and mixed road noise, which are closely associated with increased stress (*Campo, Gil & Da, 2005*; *Houshmand et al., 2012*; *Sugiharto, 2022*), reduced growth (*von Eugen et al., 2019*; *Goo et al., 2019*; *Hanafi et al., 2023*), impaired health (*Flores et al., 2019*; *Shehata et al., 2022*), and compromised production performance (*Brouček, 2014*; *Ciborowska, Michalczuk & Bień, 2021*). Designing alternative strategies to optimize the cage-housing system can provide a practical solution to ameliorate these adverse effects, meet the requirements of mitigating high stocking density- and noise-associated animal welfare, and satisfy the increasing consumers' demands for products obtained from less-stressed animals.

In chickens, adrenocorticotropic hormone (ACTH) and corticosterone (CORT) are two hypothalamus-pituitary-adrenal (HPA) axis hormones playing critical roles in stress responses (*Cheng & Muir, 2004*). Increased circulating levels of ACTH and CORT are commonly measured as stress indicators (*Ralph et al., 2015*; *Weimer et al., 2018*; *Bi et al., 2022*; *Wang et al., 2023*). Accumulated reports showed that high stocking density (*Onbaşılar & Aksoy, 2005*; *Kang et al., 2016*; *Yu et al., 2021*) and noise exposure (*Chloupek et al., 2009*) were two environmental factors capable of increasing circulating CORT level, while low stocking density (*Shehata et al., 2022*) and music auditory enhancement (*Dávila et al., 2011*) were associated with less stressful conditions. Since high stocking density and noise represent challenges and force animals to compete for limited food resources and spatial room, it is predictable that abnormal behaviors due to stress can increase as result of crowding density and environmental noise.

Contrary to noise, music has been proven to effectively enrich the environment of livestock and poultry (*Alworth & Buerkle, 2013*) and improve animal welfare by masking potentially disturbing background noises, decreasing anxiety, stress, and abnormal behaviors, and providing auditory enrichment (*Patterson-kane & Farnworth, 2006*; *Papoutsoglou et al., 2007*; *Rickard, 2009*).

On the other hand, the microbes in chicken gastrointestinal tract (GIT) play important roles in nutrient digestion and absorption, pathogen control, and immune modulation (*Oakley et al., 2014*). Harboring the highest microbial concentration and having the longest residence time of digesta in GIT, the cecum in chicken is the main site where extensive bacterial fermentation of non-starch polysaccaharides (NSPs) occurs and the main chamber in which short chain fatty acids (SCFAs) are produced (*Józefiak, Rutkowski & Martin, 2004*). SCFAs (butyric and propionic acids, particularly) play important roles in

maintaining healthy gut physiology and improving productive performance in chickens in such terms as body weight, feed conversion ratio, and meat quality (*Scicutella et al., 2021*). Therefore, the enrichment of SCFAs-producing bacterial communities in the ceca may be potential biomarkers reflecting the intestinal health status of chickens. It was reported that stress was related to the change of gut microbiota, and gut microbiota played an important role in the effects of HPA axis hormones and behaviors by a microbiota gut–brain axis (*Bienenstock, Kunze & Forsythe, 2015*; *Kraimi et al., 2019*).

High stocking density and environmental noise are common stress factors that adversely affect the health and performance of poultry, while low density and music may play the opposite roles. Stress hormone level and behaviors are commonly used to estimate animal welfare (*Costa et al., 2012*; *Johnson et al., 2019*). Changes in gut microbiota of animals, especially changes in the proprortion of beneficial flora, reflect their intestinal health. Animals with better health and welfare status may have better product quality, which is also beneficial for consumers. Yet relatively little is currently known about whether sound sources and stocking densities, particularly in combination, have any impacts on the composition and functions of gut bacterial communities in layers.

Here, this study was conducted to investigate how different (low, medium, and high) stocking densities and exposure of different sounds (natural sound, instrumental music, and traffic noise) could potentially influence physiological stress (serum ACTH and CORT levels), behaviors (preening, and feather pecking), and the cecal bacterial communities in Xiandao green-shell layers, a Chinese indigenous chicken breed with excellent egg laying performance and with increasing importance in China. Xiandao green-shell layers has good adaptability and can be reared in both the north and the south in China (*Zhou et al., 2021*). According to Chinese consumption habits, green-shelled eggs are more popular than the white-shelled ones (*Xu et al., 2024*). Furthermore, due to higher vitamins, trace elements, amino acids, and lower cholesterol and fat, green-shell eggs are considered to be an ideal natural health food (*Li et al., 2016*). We hypothesized that low stocking density and music exposure may be helpful in lowering serum stress markers, reducing stress-related behaviors, and likely benefiting intestinal microbiota in laying hens. In addition, we hope to provide evidence whether reduced product quality due to high stocking densities could be improved by music play without compromising the health status and welfare of layers.

# MATERIALS AND METHODS

## Ethics

The handling of animals in this experiment was performed in compliance with the Chinese guidelines for animal welfare and approved by the Animal Care and Use Committee of Yangzhou University (No. 202104501), Jiangsu Province, PR China.

## Animals and housing condition

This study was carried out on a commercial farm, Heheng Xiandao Poultry Co., Ltd. (Jiangyan District, Taizhou City, Jiangsu Province, China) (License number: GB14925-2010). The farm followed the principle of "all-in, all-out". Field experiments were

approved by the Animal Care and Use Committee of Yangzhou University (Approval number: 202104501). A total of 378 healthy 30-week-old Xiandao green-shell laying hens (*Gallus gallus domesticus*) were used for this experiment. These laying hens were were raised under identical feeding and management conditions. Before the experiment began, they were housed in standard commercial cages (measuring 38.5-cm long × 36.5-cm wide × 36-cm high) with three birds per cage, yielding a stocking density of approximately 468 $cm^2$ per bird. For the purpose of the current experiment, one hen was selected from each cage based on their comparable body weights, which were within the range of approximately 1.4 to 1.6 kg.

Layers were housed in three individual henhouses (7.1 m × 2.7 m × 4 m, length × width × height) under semi-controlled environmental conditions. These are three independent henhouses, separated by a 19-meter-wide greenbelt to minimize potential sound interference between them. All other facilities and conditions in these three henhouses were the same. The commercial formula diet used in this study, a corn and soybean basal feed in pellet form with no antibiotic supplementation, was produced by the Heheng Xiandao Poultry Company to meet the nutrient recommendations for laying hens according to the Nutrition Requirement of Chicken in China (NY/T 33-2004). The composition and the nutritional levels of the basal diet are presented in Table S1. Throughout the experimental period, layers had free access to feed and water. The lighting regime was 16 h of light and 8 h of darkness per day. Provision of feed occurred twice daily (7:00 in the morning and 15:00 in the afternoon). The average temperature and relative humidity of the room were 20 ± 3 °C and 65–70%, respectively. The experiment lasted 24 days after a 1-week acclimatization period.

**Experimental design**

Based on a 3 × 3 factorial arrangement with three levels of sound sources and three rates of stocking densities, layers were randomly assigned to nine groups. Layers in three henhouses were exposed to natural sound (NS), instrumental music sound (IMS), and mixed road sound (MRS), respectively. NS was the ambient sound of the henhouse. IMS was Mozart's string quartets (*Dávila et al., 2011*). MRS was a recording of traffic noise around Heheng Xiandao Poultry Co., Ltd. Music and mixed road sound were played at 65–75 dB using SAST T-50 radio twice a day from 8:00 to 11:00 in the morning and from 13:00 to 16:00 in the afternoon.

The layers were kept in galvanized-wire cages (70-cm long × 51-cm wide × 61-cm high, resulting in a floor area of 3,570 $cm^2$). The setting of different stocking densities in this experiment was based on industry practice. Currently, the conventional stocking density applied in intensified chicken farms in China is 450–500 $cm^2$/bird. Accordingly, the stocking densities used in this study were set at low density (LD), medium density (MD), and high density (HD) corresponding to four birds/cage (892 $cm^2$/bird), seven birds/cage (510 $cm^2$/bird), and 10 birds/cage (357 $cm^2$/bird), respectively.

According to the combinations of different sound sources and stocking densities, there were nine treatment groups designated as NL (NS + LD), NM (NS + MD), NH (NS + HD), IML (IMS + LD), IMM (IMS + MD), IMH (IMS + HD), MRL (MRS + LD), MRM (MRS +

MD), and MRH (MRS + HD). Each group consisted of six replicates. There were 18 cages in each henhouse, totaling 126 layers per room, and the 18 cages were arranged in six rows randomly in each henhouse (Fig. S1). The radios were placed in the center of the henhouses and raised with brackets to ensure that the difference in decibels of indoor rooms does not exceed 10 dB.

## Sample collection and analysis

On the 3rd, 12th, and 24th experimental day (D3, D12 and D24, respectively), blood samples were collected from six randomly selected birds (one layer per replicate) ($n = 6$). Briefly, sub-wing vein blood (about 5 mL for each hen) was collected through venipuncture using non-EDTA treated vacutainer tubes (Becton Dickinson, Franklin Lakes, NJ, USA). The blood sample was allowed to clot at room temperature for half an hour before the serum was harvested by a centrifugation step (3,000× $g$ for 30 min at 4 °C). The concentrations of ACTH and CORT in the serum were measured using commercial enzyme-linked immunosorbent assay (ELISA) kits (AMOKE, Lianshuo Biotechnology Co., Ltd., Shanghai, China) according to the instructions of the manufacturer. Standard curves constructed for each of the assayed hormones had a regression value higher than 0.99. The intra- and inter-assay coefficients of variation were <9 and 11% for ACTH and CORT.

Observation of the behaviors including preening and feather pecking was carried out using focal animal sampling method. Preening was recorded when a layer uses its beak to peck, stroke or comb plumage (Ventura, Siewerdt & Estevez, 2012). Feather pecking means the layer pecks towards the feathers of the others in cage (Li et al., 2019). Representative images explaining the two specific behaviors were shown in Fig. S2, Videos S1 and S2. Briefly, an overhead camera system was used to record the behavior of layers on D3, D12 and D24 for 15 min (from 9:00 to 9:15). Fifteen birds per group (two or three layers from each replicate) were randomly selected for behavior evaluation ($n = 15$). These selected birds were differentiated by their feet marks (a foot ring made by plastics tied on the leg). The recorded behavior of these selected and labeled birds were analyzed using Noldus Observer XT (Version 7.0, Noldus Information Technology BV, Wageningen, Netherland) and by the same observer who was blind to the treatments. When the observer observed the target behavior, he used the software (Noldus Observer XT) to record it. The end of one target behavior was determined by observing a 4-s interval after the behavior stopped or when another behavior was performed. Behavior frequency (number, $n$) and behavior duration (second, s) were defined as the number of times and the total duration, respectively, of the observed behavior occurred per individual bird during the observation period.

At the end of the experiment, all layers underwent an overnight fasting and six randomly selected birds (one layer per replicate) were euthanized by cervical dislocation ($n = 6$). The ceca were removed from the gastrointestinal tract using sterile scissors. Cecal contents from both ceca were carefully hand-stripped into aseptic plastic tubes, immersed in liquid nitrogen immediately, and stored at −80 °C for subsequent 16S rRNA amplicon sequencing analysis. After the test, the other laying hens remained on Heheng Xiandao

Poultry Co., Ltd. (Jiangyan District, Taizhou City, Jiangsu Province, China) for breeding and production purposes.

To assess the composition of cecal microbiota in different treatment groups, cecal contents (around 0.5 g per sample) were used for amplification and sequencing of 16S rRNA V4–V5 hypervariable region, which was completed by staff in Genepioneer Biotechnologies Co. Ltd., Nangjing, China. Briefly, the total microbial community DNA in the cecal contents was extracted using the QIAamp DNA Stool Mini Kit (Qiagen, Hilden, Germany) according to the producer's instructions. The quantity and quality of microbial genomic DNA were examined spectrophotometrically using NanoDrop ND-2000 spectrophotometer (Thermo Fisher Scientific, Waltham, MA, USA) and 1.5% agarose gel electrophoresis. The V4–V5 region of the 16S rRNA gene was amplified with a primer set of 515F (5′-GTGCCAGCMGCCGCGG-3′) and 907R (5′-CCGTCAATTCMTTTRAG TTT-3′). Details regarding the composition of the PCR system, thermocycling conditions, and ligation of PCR products with unique index were performed according to previous descriptions (*Fadrosh et al., 2014*). The obtained PCR products were purified using the AxyPrep DNA Gel Extraction Kit (Axygen Biosciences, Union City, CA, USA) and quantified using Qubit fluorometer (Invitrogen, Carlsbad, CA, USA). Equimolar amounts of differently barcoded 16S rRNA gene amplicons from each sample were pooled and sequenced on the Illumina Miseq platform (Illumina, San Diego, CA, USA).

The obtained reads were assigned to different samples according to the barcodes. Low quality sequence reads were removed if the length was shorter than 200 bp, had an average phred score of lower than 20, contained ambiguous bases, or contained homopolymers of >9 bp. Sequences passed the quality filtering were merged using FLASH v1.2.7 (https:// sourceforge.net/projects/flashpage/files/). The chimeric sequences were identified by comparing with the reference database (Gold Database) using UCHIME algorithm and removed using the USEARCH 6.1 database (*Edgar, 2013*). The resulting effective sequences were clustered into operational taxonomic units (OTUs) at an identity threshold of 97% similarity by calling uclust from Qiime (http://qiime.org/scripts/pick_otus.html) (*Caporaso et al., 2010*). The rarefy function in QIIME pipeline was applied to avoid the impacts of different sequencing depth on the measurement of bacterial compositions. For each OTU, the longest sequence in each cluster was selected as the representative and used to annotate taxonomic information using the RDP Classifier (http://sourceforge.net/ projects/rdp-classifier) and the Silva 16S sequence database.

Both alpha- and beta-diversity indices were calculated by QIIME software (version 1.9.1) and used to estimate the microbiome diversity within and between bacterial communities. Nonmetric multidimensional scaling (NMDS) graphs were constructed using Bray-Curtis dissimilarity values. Analysis of variance of permutation (PERMANOVA) from R's package Vegan was used to compare the effects of sound and density on community structures. LEfSe algorithm was used to carry out linear discriminant analysis (LDA) on samples according to the taxonomic compositions (*Segata et al., 2011*). Differences with LDA score >4.0 and $P$ value < 0.05 were considered statistically significant.

Functional capabilities of cecal bacterial communities were predicted through PICRUSt (http://picrust.github.io/picrust) according to the abundance of OTU level (*Langille et al., 2013*). The predicted genes and their function were aligned to the Kyoto Encyclopedia of Genes and Genomes (KEGG) pathways and the differences among groups were compared using STAMP (http://kiwi.cs.dal.ca/Software/STAMP). Two-side Welch's t-test and Benjamini-Hochberg FDR correction were employed.

## Statistical analysis

The experiment was set up as a completely randomized design. Data obtained from this study were statistically analyzed using SPSS 22.0 for Windows (SPSS, Inc., Chicago, IL, USA). Prior to analysis, normal distribution of data as well as the independence and homogeneity of variances among treatment groups was verified. For serum measurements, data on different days were subjected to two-way ANOVA in a $3 \times 3$ factorial arrangement with three levels of stocking density (LD, MD and HD) and three different sound sources (NS, IMS and MRS) to analyze the main effects of sound and stocking density alone, as well as their interactions. When there were interactions, simple effect analysis was performed. Duncan's test was used for *post-hoc* multiple comparisons. Behavior data on different days were analyzed by General Linear Mixed Models (GLMM) with sound source and stocking density as fixed factors, and individuals as a random factor. The models consisted of behaviors of the frequency and duration of preening and feather pecking ($Freq_{preen}$, $Freq_{peck}$, $Dura_{preen}$ and $Dura_{peck}$, respectively) as target variables, sound source, stocking density and their interaction as the fixed effects. We established poisson distribution error with log link function for $Freq_{preen}$ and $Freq_{peck}$, and normal distribution error with identity link function for $Dura_{preen}$ and $Dura_{peck}$. *Post-hoc* pairwise comparisons were adjusted by Bonferroni correction. For bioinformatics analysis, Alpha diversity indices (observed species, Chao 1, good's coverge, PD whole tree, Shannon and Simpson index), Beta diversity analysis of Bray-Curtis distance, and the relative abundance of bacteria were calculated using QIIME software. NMDS was chosen to visualize the dissimilarity matrix between samples which was computed through Bray-Curtis distance for Beta diversity. Differentially abundant taxa were identified by the linear discriminant analysis (LDA) effect size (LEfSe) analysis (http://www.ehbio.com/Cloud_Platform/front/). The prediction and annotation of biological functions of cecal bacteria were performed using KEGG pathway and PICRUSt. Furthermore, differences between the two groups in taxonomic and diversity data were compared with Welch's two-sided t-test. For multigroup comparison, two-way analyses of variance (ANOVA) in combination with Kruskal-Wallis test were applied. Statistical analysis was performed using the transformed data, while means from raw data were used for presentation. Data are presented as mean ± SEM. A *P* value less than 0.05 was considered to be significant.

## RESULTS

### Serum levels of ACTH and CORT

At three sampling time points, no significant interactions between sound source and stocking density on the serum levels of ACTH and CORT were observed. Significant main

**Table 1 Serum levels of stress-related hormones at different sampling points ($n$ = 6).**

| | | ACTH (ng/L) | | | CORT (ng/mL) | | |
|---|---|---|---|---|---|---|---|
| | | Day 3 | Day 12 | Day 24 | Day 3 | Day 12 | Day 24 |
| Group | NL | 11.28 ± 1.34 | 4.22 ± 0.82 | 4.03 ± 1.55 | 17.81 ± 2.90 | 6.20 ± 1.21 | 16.07 ± 2.22 |
| | NM | 15.47 ± 2.24 | 7.69 ± 1.46 | 3.87 ± 0.44 | 17.92 ± 2.15 | 9.45 ± 1.00 | 18.02 ± 2.47 |
| | NH | 11.72 ± 1.90 | 11.00 ± 0.98 | 10.39 ± 0.45 | 19.44 ± 1.49 | 8.68 ± 1.08 | 23.92 ± 5.19 |
| | IML | 11.14 ± 0.87 | 3.18 ± 0.54 | 3.02 ± 0.97 | 16.30 ± 1.57 | 5.24 ± 2.08 | 11.83 ± 1.46 |
| | IMM | 10.88 ± 1.19 | 5.70 ± 0.18 | 2.44 ± 0.61 | 17.90 ± 1.83 | 6.46 ± 1.00 | 17.23 ± 2.64 |
| | IMH | 10.83 ± 1.06 | 10.77 ± 1.93 | 10.10 ± 1.76 | 16.93 ± 2.92 | 12.81 ± 2.18 | 23.75 ± 3.29 |
| | MRL | 19.41 ± 1.78 | 4.27 ± 0.40 | 4.05 ± 0.55 | 24.00 ± 2.47 | 5.56 ± 0.86 | 16.91 ± 2.17 |
| | MRM | 16.37 ± 1.39 | 7.88 ± 1.32 | 2.90 ± 0.70 | 18.22 ± 1.89 | 9.56 ± 1.55 | 23.44 ± 3.10 |
| | MRH | 21.44 ± 1.15 | 11.36 ± 2.13 | 9.64 ± 2.35 | 24.81 ± 3.74 | 14.88 ± 2.63 | 23.87 ± 1.42 |
| Main effect | | | | | | | |
| Sound (S) | NS | 12.82 ± 1.11[B] | 7.64 ± 0.91 | 6.10 ± 0.90 | 18.39 ± 1.24[ab] | 8.11 ± 0.69 | 19.34 ± 2.09 |
| | IMS | 10.95 ± 0.57[B] | 6.55 ± 0.99 | 5.19 ± 1.07 | 17.04 ± 1.20[b] | 8.17 ± 1.28 | 17.60 ± 1.83 |
| | MRS | 19.07 ± 0.94[A] | 7.83 ± 1.06 | 5.53 ± 1.06 | 22.35 ± 1.68[a] | 10.00 ± 1.36 | 21.41 ± 1.48 |
| Density (D) | LD | 13.94 ± 1.20 | 3.89 ± 0.35[C] | 3.70 ± 0.61[B] | 19.37 ± 1.52 | 5.67 ± 0.81[C] | 14.94 ± 1.20[B] |
| | MD | 14.24 ± 1.08 | 7.09 ± 0.67[B] | 3.07 ± 0.35[B] | 18.01 ± 1.06 | 8.49 ± 0.74[B] | 19.56 ± 1.63[AB] |
| | HD | 14.66 ± 1.40 | 11.04 ± 0.95[A] | 10.05 ± 0.93[A] | 20.40 ± 1.75 | 12.12 ± 1.28A | 23.84 ± 1.98[A] |
| P value | | | | | | | |
| Sound | | <0.001 | 0.411 | 0.657 | 0.029 | 0.281 | 0.277 |
| Density | | 0.839 | <0.001 | <0.001 | 0.490 | <0.001 | 0.002 |
| S × D | | 0.051 | 0.960 | 0.971 | 0.499 | 0.133 | 0.751 |

**Notes:**
ACTH, adrenocorticotropic hormone; CORT, corticosterone; NS, natural sound; IMS, instrumental music; MRS, mixed road sound; LD, low density; MD, medium density; HD, high density; NL, NS + LD; NM, NS + MD; NH, NS + HD; IML, IMS + LD; IMM, IMS + MD; IMH, IMS + HD; MRL, MRS + LD; MRM, MRS + MD; MRH, MRS + HD. Data are presented as mean ± standard error of the mean (SEM).
[a,b] Means with different low case letters within a column indicate significant differences ($P \leq 0.05$).
[A,B,C] Means with different capital letters within a column indicate very significant differences ($P \leq 0.01$).

effects of sound source on serum ACTH ($P_{(sound)}$ < 0.001) and CORT ($P_{(sound)}$ = 0.029) concentrations were only observed at the early stage of this experiment (D3) with layers in group MRS having the highest levels. Significant main effects of stocking density on serum ACTH and CORT levels were observed at D12 and D24 ($P_{(density)} \leq 0.002$). Layers kept at HD were characterized by significantly higher ACTH and CORT levels than layers kept at LD (Table 1).

Our results showed that IMS and LD were associated with reduced levels of two stress-related hormones, while MRS and HD could induce elevated ACTH and CORT levels.

## Behavior observation

### The frequency and duration of preening behaviors

On D3, significant S × D (sound and density) interactions were observed on the preening behaviors. Under LD and HD, IMS-exposed layers had significantly increased $Freq_{preen}$ ($P \leq 0.017$) and significantly longer $Dura_{preen}$ ($P \leq 0.019$) when compared with

**Table 2 The frequency (n) and duration (s) of preening behavior ($n = 15$).**

| | | Frequency | | | Duration | | |
|---|---|---|---|---|---|---|---|
| | | Day 3 | Day 12 | Day 24 | Day 3 | Day 12 | Day 24 |
| Group | NL | $2.67 \pm 0.64^{bd}$ | $3.93 \pm 0.82$ | $6.80 \pm 0.73$ | $33.73 \pm 13.91^{c}$ | $63.33 \pm 14.85$ | $63.80 \pm 9.48$ |
| | NM | $2.87 \pm 0.66^{b}$ | $4.93 \pm 0.61$ | $5.47 \pm 0.75$ | $23.87 \pm 9.24^{bc}$ | $61.87 \pm 14.17$ | $35.33 \pm 5.44$ |
| | NH | $3.80 \pm 0.44^{bc}$ | $6.00 \pm 1.04$ | $5.40 \pm 0.75$ | $34.80 \pm 5.36^{bc}$ | $66.40 \pm 17.00$ | $33.33 \pm 7.25$ |
| | IML | $9.67 \pm 1.47^{a}$ | $7.60 \pm 0.89$ | $13.47 \pm 0.99$ | $158.13 \pm 28.82^{a}$ | $87.93 \pm 20.77$ | $84.93 \pm 8.85$ |
| | IMM | $4.73 \pm 0.74^{b}$ | $6.67 \pm 1.04$ | $9.33 \pm 1.09$ | $43.33 \pm 10.15^{b}$ | $92.67 \pm 23.71$ | $78.07 \pm 14.21$ |
| | IMH | $5.80 \pm 0.33^{ab}$ | $6.73 \pm 0.89$ | $11.53 \pm 1.32$ | $67.07 \pm 8.52^{b}$ | $80.60 \pm 13.60$ | $74.73 \pm 14.08$ |
| | MRL | $1.53 \pm 0.52^{d}$ | $3.33 \pm 0.91$ | $3.67 \pm 0.95$ | $10.67 \pm 5.14^{c}$ | $45.07 \pm 15.28$ | $23.67 \pm 8.59$ |
| | MRM | $3.53 \pm 0.79^{bc}$ | $2.20 \pm 0.37$ | $4.93 \pm 0.92$ | $26.27 \pm 6.84^{bc}$ | $19.00 \pm 4.14$ | $17.27 \pm 4.03$ |
| | MRH | $2.87 \pm 0.47^{cd}$ | $3.40 \pm 0.55$ | $4.27 \pm 0.67$ | $24.93 \pm 5.75^{c}$ | $33.40 \pm 7.14$ | $18.47 \pm 4.58$ |
| Main effect | | | | | | | |
| Sound (S) | NS | $3.11 \pm 0.34^{B}$ | $4.96 \pm 0.49^{B}$ | $5.89 \pm 0.43^{B}$ | $30.80 \pm 5.76^{B}$ | $63.87 \pm 8.68^{A}$ | $44.16 \pm 4.76^{B}$ |
| | IMS | $6.73 \pm 0.63^{A}$ | $7.00 \pm 0.54^{A}$ | $11.44 \pm 0.69^{A}$ | $89.51 \pm 12.74^{A}$ | $87.07 \pm 11.20^{A}$ | $79.24 \pm 7.15^{A}$ |
| | MRS | $2.64 \pm 0.37^{B}$ | $2.98 \pm 0.38^{C}$ | $4.29 \pm 0.49^{C}$ | $20.62 \pm 3.52^{B}$ | $32.49 \pm 5.88^{B}$ | $19.80 \pm 3.46^{C}$ |
| Density (D) | LD | $4.62 \pm 0.77$ | $4.96 \pm 0.57$ | $7.98 \pm 0.80$ | $67.51 \pm 14.38^{A}$ | $65.44 \pm 10.04$ | $57.47 \pm 6.35$ |
| | MD | $3.71 \pm 0.43$ | $4.60 \pm 0.49$ | $6.58 \pm 0.60$ | $31.16 \pm 5.16^{B}$ | $57.84 \pm 10.17$ | $43.56 \pm 6.41$ |
| | HD | $4.16 \pm 0.30$ | $5.38 \pm 0.53$ | $7.07 \pm 0.72$ | $42.27 \pm 4.65^{B}$ | $60.13 \pm 8.03$ | $42.18 \pm 6.46$ |
| *P* value | | | | | | | |
| Sound | | <0.001 | <0.001 | <0.001 | <0.001 | <0.001 | <0.001 |
| Density | | 0.471 | 0.309 | 0.693 | 0.002 | 0.829 | 0.083 |
| S × D | | 0.001 | 0.268 | 0.241 | <0.001 | 0.848 | 0.609 |

**Notes:**
NS, natural sound; IMS, instrumental music; MRS, mixed road sound; LD, low density; MD, medium density; HD, high density; NL, NS + LD; NM, NS + MD; NH, NS + HD; IML, IMS + LD; IMM, IMS + MD; IMH, IMS + HD; MRL, MRS + LD; MRM, MRS + MD; MRH, MRS + HD. Data are presented as mean ± standard error of the mean (SEM).
[a,b,c,d] Means with different low case letters within a column indicate significant differences ($P \leq 0.05$).
[A,B,C] Means with different capital letters within a column indicate very significant differences ($P \leq 0.01$).

MRS-exposed layers. Among three IMS-exposed groups, group IML had significantly higher $Freq_{preen}$ ($P = 0.008$) and $Dura_{preen}$ ($P < 0.001$) than group IMM.

The main effects of sound on $Freq_{preen}$ and $Dura_{preen}$ were observed on D12 and D24. MRS-exposed groups was always associated with significantly lower $Freq_{preen}$ ($P_{(sound)} \leq 0.017$) and $Dura_{preen}$ ($P_{(sound)} \leq 0.015$) when compared with NS- and IMS-exposed groups on D12 and D24. IMS-exposed groups were characterized by significantly higher $Freq_{preen}$ ($P_{(sound)} < 0.001$) and $Dura_{preen}$ ($P_{(sound)} < 0.001$) than NS-exposed groups on D24 (Table 2).

### The frequency and duration of feather pecking behavior
Our results showed that, there were significant S × D interactions on $Freq_{peck}$ on D3. Regardless of stocking densities, IMS-exposed groups always had significantly lower $Freq_{peck}$ than MRS-exposed groups ($P < 0.001$) on D3, and MRS was always associated with significantly higher $Freq_{peck}$ when compared with NS and IMS ($P \leq 0.039$). Among three NS-exposed groups, group NL was associated with significantly lower $Freq_{peck}$ than

**Table 3 The frequency (n) and duration (s) of feather pecking behavior ($n = 15$).**

| | | Frequency | | | Duration | | |
|---|---|---|---|---|---|---|---|
| | | Day 3 | Day 12 | Day 24 | Day 3 | Day 12 | Day 24 |
| Group | NL | $0.73 \pm 0.18^c$ | $1.80 \pm 0.33$ | $2.40 \pm 0.53$ | $2.07 \pm 0.54$ | $6.93 \pm 1.54$ | $8.20 \pm 2.61$ |
| | NM | $2.93 \pm 0.60^b$ | $4.53 \pm 0.60$ | $5.00 \pm 0.78$ | $13.07 \pm 3.85$ | $22.60 \pm 3.99$ | $22.07 \pm 4.26$ |
| | NH | $2.67 \pm 0.45^b$ | $6.00 \pm 0.89$ | $5.27 \pm 0.76$ | $12.73 \pm 2.70$ | $31.00 \pm 6.45$ | $30.27 \pm 11.16$ |
| | IML | $0.53 \pm 0.19^c$ | $0.80 \pm 0.24$ | $1.40 \pm 0.38$ | $1.80 \pm 0.63$ | $2.33 \pm 0.78$ | $3.13 \pm 1.04$ |
| | IMM | $0.67 \pm 0.30^c$ | $0.87 \pm 0.53$ | $1.73 \pm 0.34$ | $1.87 \pm 0.82$ | $4.73 \pm 3.01$ | $4.47 \pm 0.81$ |
| | IMH | $0.40 \pm 0.16^c$ | $1.07 \pm 0.37$ | $2.67 \pm 0.40$ | $1.47 \pm 0.60$ | $7.60 \pm 3.65$ | $7.33 \pm 1.45$ |
| | MRL | $4.93 \pm 0.80^a$ | $5.07 \pm 0.73$ | $7.33 \pm 0.89$ | $22.07 \pm 5.97$ | $29.87 \pm 5.33$ | $26.80 \pm 5.73$ |
| | MRM | $4.67 \pm 0.50^a$ | $6.13 \pm 1.01$ | $12.40 \pm 1.00$ | $25.87 \pm 3.93$ | $30.60 \pm 4.22$ | $50.13 \pm 6.21$ |
| | MRH | $5.93 \pm 0.87^a$ | $5.47 \pm 0.75$ | $10.87 \pm 1.08$ | $36.47 \pm 7.39$ | $40.80 \pm 6.33$ | $37.20 \pm 4.49$ |
| Main effect | | | | | | | |
| Sound (S) | NS | $2.11 \pm 0.29^B$ | $4.11 \pm 0.45^B$ | $4.22 \pm 0.44^B$ | $9.29 \pm 1.72^B$ | $20.18 \pm 2.94^B$ | $20.18 \pm 4.21^B$ |
| | IMS | $0.53 \pm 0.13^C$ | $0.91 \pm 0.23^C$ | $1.93 \pm 0.23^C$ | $1.71 \pm 0.39^C$ | $4.89 \pm 1.60^C$ | $4.98 \pm 0.69^C$ |
| | MRS | $5.18 \pm 0.43^A$ | $5.56 \pm 0.48^A$ | $10.20 \pm 0.65^A$ | $28.13 \pm 3.47^A$ | $33.76 \pm 3.12^A$ | $38.04 \pm 3.43^A$ |
| Density (D) | LD | $2.07 \pm 0.41^b$ | $2.56 \pm 0.39$ | $3.71 \pm 0.53^B$ | $8.64 \pm 2.43^b$ | $13.04 \pm 2.57^B$ | $12.71 \pm 2.58^B$ |
| | MD | $2.76 \pm 0.37^{ab}$ | $3.84 \pm 0.54$ | $6.38 \pm 0.80^A$ | $13.60 \pm 2.34^{ab}$ | $19.31 \pm 2.68^B$ | $25.56 \pm 3.76^A$ |
| | HD | $3.00 \pm 0.47^a$ | $4.18 \pm 0.52$ | $6.27 \pm 0.68^A$ | $16.89 \pm 3.38^a$ | $26.47 \pm 3.81^A$ | $24.93 \pm 4.39^A$ |
| $P$ value | | | | | | | |
| Sound | | <0.001 | <0.001 | <0.001 | <0.001 | <0.001 | <0.001 |
| Density | | 0.043 | 0.056 | <0.001 | 0.030 | 0.001 | 0.004 |
| S × D | | 0.009 | 0.066 | 0.313 | 0.206 | 0.193 | 0.100 |

**Notes:**

NS, natural sound; IMS, instrumental music; MRS, mixed road sound; LD, low density; MD, medium density; HD, high density; NL, NS + LD; NM, NS + MD; NH, NS + HD; IML, IMS + LD; IMM, IMS + MD; IMH, IMS + HD; MRL, MRS + LD; MRM, MRS + MD; MRH, MRS + HD. Data are presented as mean ± standard error of the mean (SEM).

[a,b,c] Means with different low case letters within a column indicate significant differences ($P \leq 0.05$).

[A,B,C] Means with different capital letters within a column indicate very significant differences ($P \leq 0.01$).

the other two groups ($P \leq 0.001$) on D3 with a pattern that the lower the density, the lower the $\text{Freq}_{\text{peck}}$.

The main effects of sound on $\text{Freq}_{\text{peck}}$ and $\text{Dura}_{\text{peck}}$ on D3, D12 and D24 were observed. IMS was also associated with significantly lower $\text{Freq}_{\text{peck}}$ ($P_{(\text{sound})} < 0.001$) and $\text{Dura}_{\text{peck}}$ ($P_{(\text{sound})} \leq 0.016$), while MRS was correlated with significantly higher $\text{Freq}_{\text{peck}}$ ($P_{(\text{sound})} \leq 0.005$) and $\text{Dura}_{\text{peck}}$ ($P_{(\text{sound})} < 0.001$).

Stocking density had significant main effects on $\text{Freq}_{\text{peck}}$ on D24 and $\text{Dura}_{\text{peck}}$ on D3, D12 and D24. Layers kept at LD always had significantly lower $\text{Freq}_{\text{peck}}$ ($P_{(\text{density})} < 0.001$) and $\text{Dura}_{\text{peck}}$ ($P_{(\text{density})} \leq 0.009$) when compared with layers kept at HD (Table 3).

## Effects of sound and stocking density on cecal bacterial communities

### General information of sequencing

In this study, a total of 3,583,439 high-quality reads were generated from 54 cecal luminal contents (an average of 66,360 reads per sample) and 1,319 OTUs were obtained covering

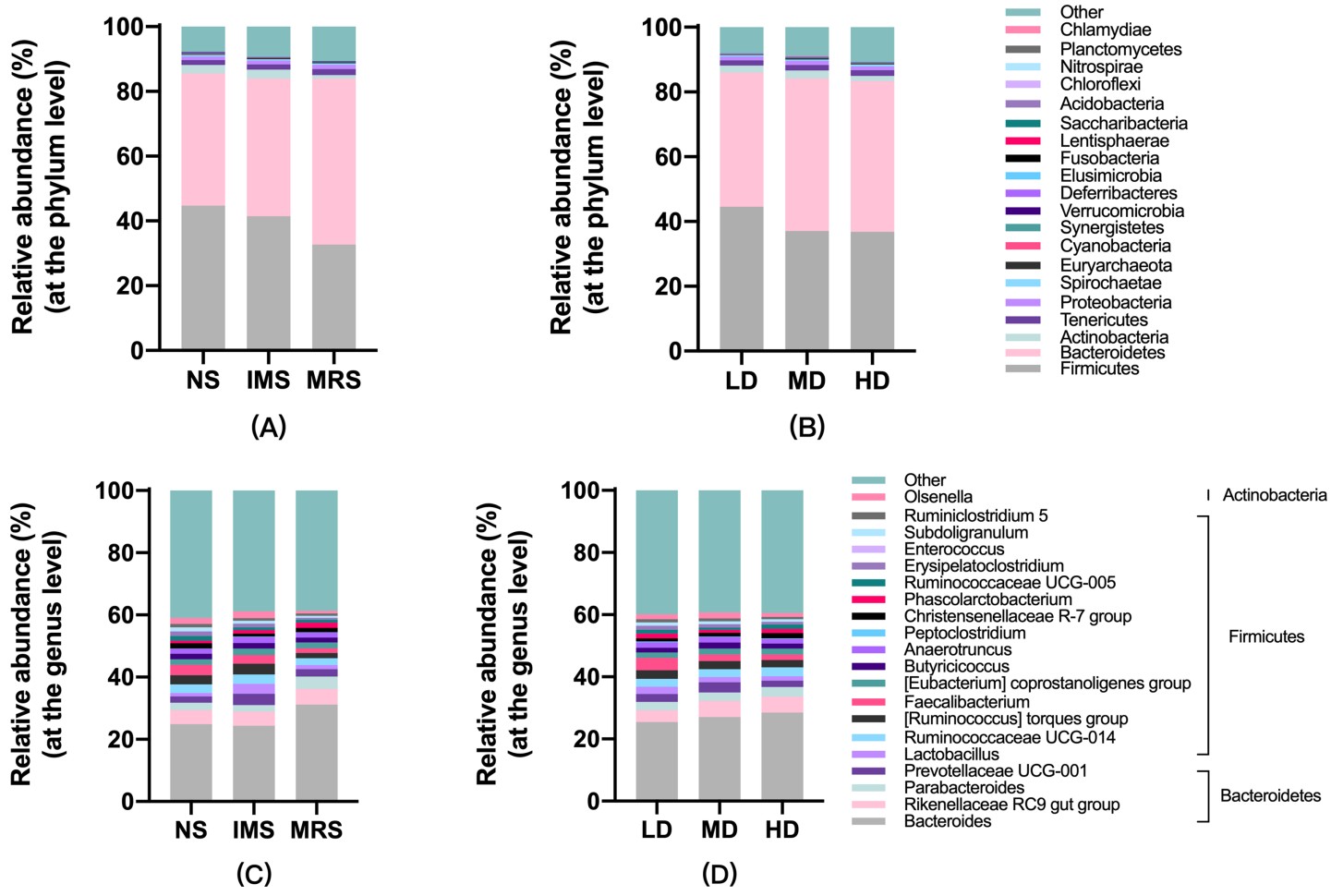

**Figure 1 Taxonomic profiles of cecal bacteria among layers exposed to different sound sources and kept at different stocking densities.** (A) and (B), at the phylum; (C) and (D), at the genus level. Abbreviations: NS, natural sound; IMS, instrumental music; MRS, mixed road sound; LD, low density; MD, medium density; HD, high density.

26 phyla. The average number of OTUs for each sample was 675 with 76 core OTUs consistently detected in all samples.

### Taxonomic composition variations in cecal bacterial communities

Assigned taxonomic profiles revealed that, across all treatment groups, Bacteroidetes and Firmicutes dominated the cecal bacterial communities with an average abundance of 45.10% and 39.40%, respectively, followed by Actinobacteria (2.10%), Tenericutes (1.67%), and Proteobacteria (1.20%). LEfSe results showed that bacteria in genus *Lactobacillus* (phylum Firmicutes, class Bacilli, order Lactobacillales, family Lactobacillaceae) and *Ruminococcus torques group* were significantly enriched in cecal contents collected from IMS-exposed layers, while the relative abundance of bacteria in genera *Bacteroides* and *Parabacteroides* was significantly increased in cecal digesta collected from MRS-exposed layers. The bacterial community structure in the ceca of NS-exposed layers was characterized by significantly enriched bacteria in genus *Faecalibacterium* ($P < 0.05$ and LDA score >4.0) (Figs. 1 and 2). Among layers kept at different stocking densities, genera

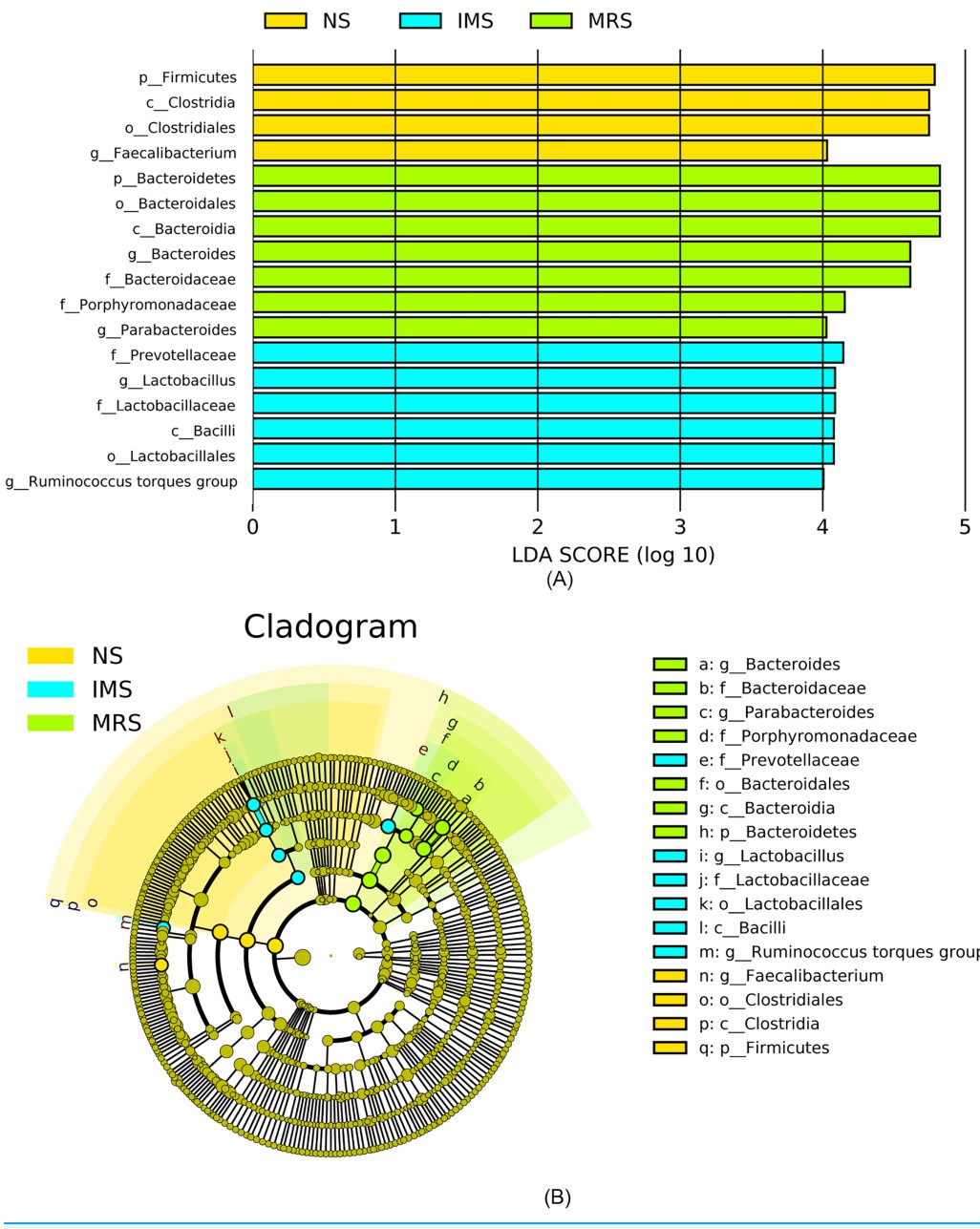

**Figure 2 Most differentially abundant taxa in the cecal digesta across different treatment groups.**
(A) Indicator bacteria identified by linear discriminant analysis (LDA) with a LDA score larger than 4. (B) Cladogram showing the phylogenetic distribution of the bacterial lineages. Circles indicate phylogenetic levels from phylum to genus and the diameter of each circle is proportional to the abundance of the group. Abbreviations: NS, natural sound; IMS, instrumental music; MRS, mixed road sound.

*Faecalibacterium* and *Megamonas* were bacterial biomarkers in the ceca of layers kept at LD, while genera *Ruminococcaceae UCG 005*, *Ruminococcaceae UCG 010*, and *Lachnospiraceae NK4A136 group* were significantly enriched in the ceca of layers kept at HD. Significantly enriched bacteria in genera *Rikenellaceae RC9 gut group* and *Odoribacter*

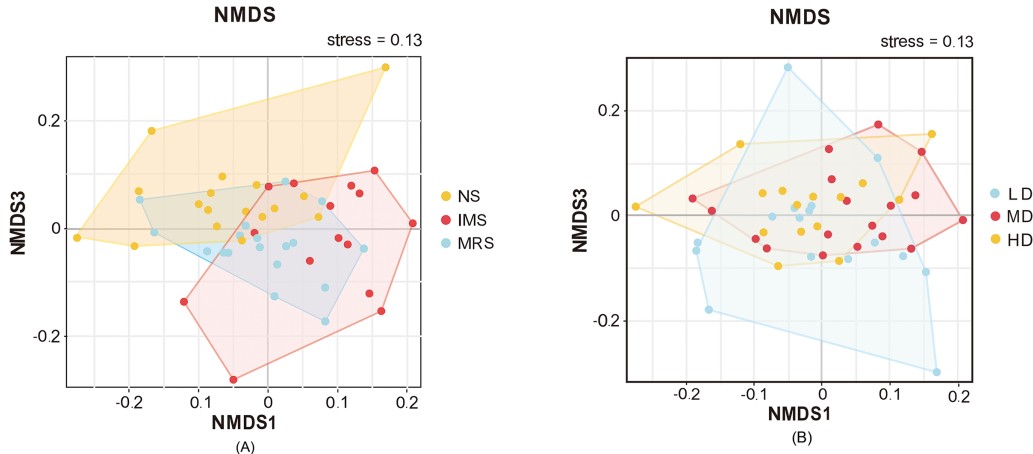

**Figure 3 Beta-diversity of the cecal microbiome illustrated by non-metric multidimensional scaling (NMDS) plots.** (A and B) NMDS plots among groups exposed to different sound sources and stocking densities, respectively. Each point represents an individual cecal microbiome. Abbreviations: NS, natural sound; IMS, instrumental music; MRS, mixed road sound; LD, low density; MD, medium density; HD, high density.

were associated with cecal contents collected from layers kept at MD ($P < 0.05$ and LDA score > 2.5) (Figs. 1 and S3).

***Alpha- and Beta-diversity of the bacterial communities in the cecal contents***
All good's coverages were larger than 0.996, which suggested that the bacterial diversity within cecal digesta samples had been sufficiently captured. Neither significant main effects of stocking density nor significant S × D interactions were observed in all the indices evaluated. Sound had significant main effects on the Shannon ($P_{(sound)} = 0.009$) and Simpson ($P_{(sound)} = 0.034$) indices with IMS-exposed layers exhibiting significantly lower values than NS- and MRS-exposed layers, which reflected a significantly less diverse bacterial community structure in the cecal digesta due to IMS exposure (Table S2). Consistently, cecal contents collected from group IMS had a relatively lower number of OTUs (737 ± 13) than that from group NS (760 ± 13) and group MRS (762 ± 10).

Furthermore, NMDS based on Bray-Curtis dissimilarities revealed the separation of samples among different groups (Fig. 3). The bacterial community compositions of cecal digesta were differentiated among layers exposed to different sound sources, whereas the separation among layers housed at different stocking density was hardly detected.

Prediction of molecular functions of the cecal luminal microbiomes was conducted using PICRUSt. The cecal microbiomes were further analyzed with PICRUSt to predict their potential functions. At KEGG level 2, cecal bacteria in NS- and IMS-exposed layers had significantly higher relative abundance of predicted gene functions involved in the environmental information processing pathway of membrane transport ($P = 0.001$), the metabolic pathways of lipid metabolism ($P = 0.044$) and xenobiotics biodegradation and metabolism ($P = 0.009$) than that in MRS-exposed layers. Meanwhile, the pathways of transport and catabolism (cellular processes) ($P < 0.001$), folding, sorting and degradation (genetic information processing) ($P = 0.001$), as well as the metabolic pathways of

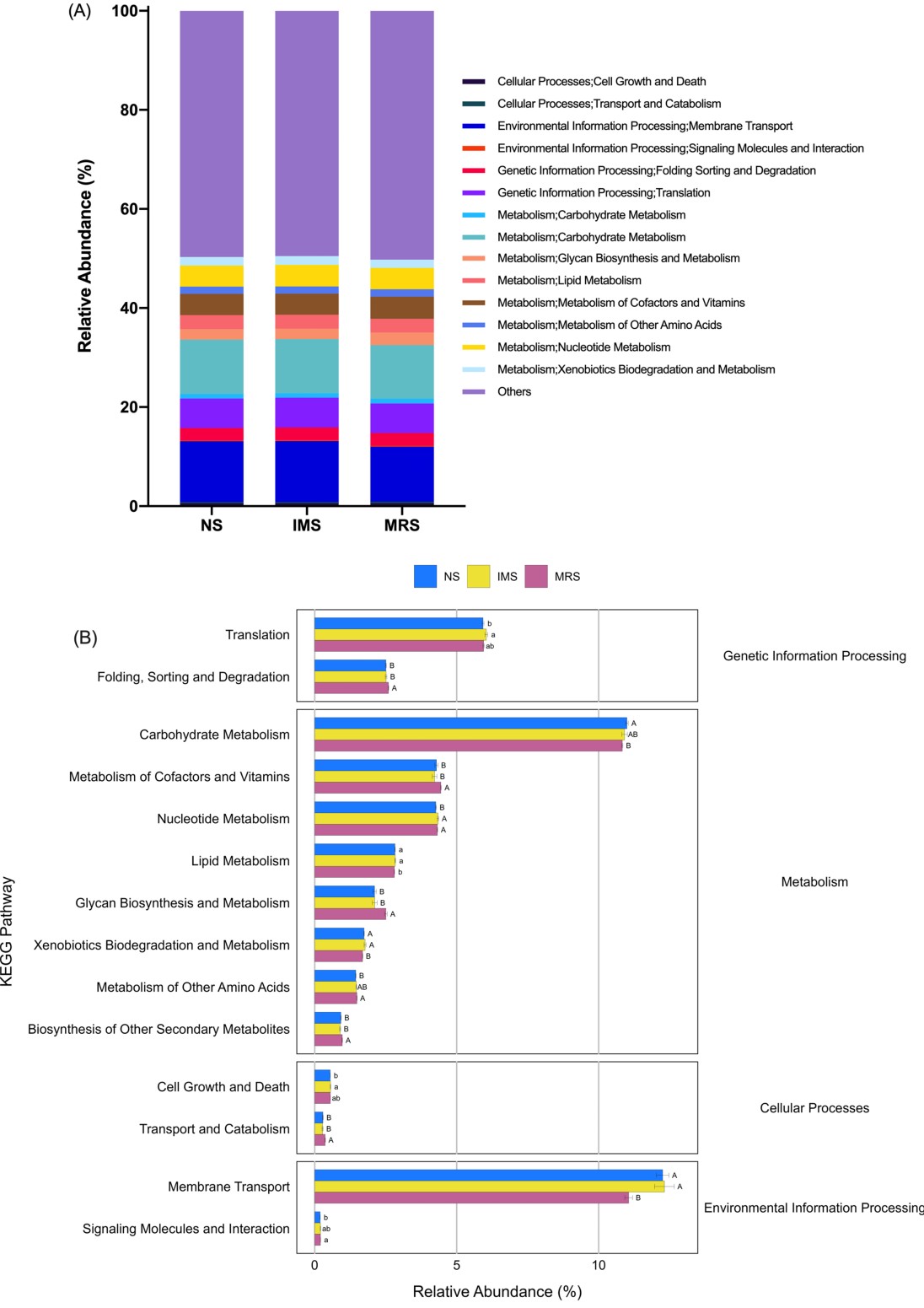

**Figure 4 Prediction of function of the cecal microbiota of green-shell laying hens across different groups.** (A) KEGG annotation at level 2 under different sound exposures; (B) Differences in the enrichment of various KEGG pathways at level 2. NS, natural sound; IMS, instrumental music; MRS, mixed road sound. Data are presented as mean ± standard error of the mean (SEM). Error bars indicate standard errors. Different low case letters and capital letters connect bar indicate significant differences ($P \leq 0.05$, $P \leq 0.01$, respectively).

biosynthesis of other secondary metabolites ($P = 0.003$), glycan biosynthesis and metabolism($P < 0.001$), metabolism of cofactors and vitamins ($P = 0.009$) were significantly enriched in MRS groups (Fig. 4). For stocking density, there was no differences in the enrichment of various KEGG pathways at level 2 in Xiandao green-shell layers (Fig. S4).

## DISCUSSION

### Impacts of different stocking densities and sounds on physiological stress

Results from the present study demonstrated the association of LD and IMS with alleviated physiological stress in Xiandao green-shell layers, evidenced by the significantly lower serum ACTH and CORT levels when compared with HD and MRS, respectively. Activation of the HPA axis and thus the increased secretion of ACTH and CORT are generally involved in stress responses and negatively influence animals' performance (*Scanes, 2016*). A number of studies have been conducted aiming to counteract the adverse influences of HD-induced stress on the productive performance of chickens. For example, dietary supplementation of organic chromium (*Mirfendereski & Jahanian, 2015*), 25-hydroxycholecalciferol (*Wang et al., 2020*), germinated paddy rice in laying hens (*Incharoen, Roytrakul & Likittrakulwong, 2021*), and alpha-lipoic acid in broilers (*Ma et al., 2020*) have been demonstrated to decrease the plasma CORT level caused by HD and thus improve animals' productive performance. In the meanwhile, exposure of layers to classical music was shown to be effective in reducing stress levels evidenced by significantly decreased heterophil to lymphocyte ratio (H/L) (*Dávila et al., 2011*). Taken together, our results suggested LD and IMS exposure may be further explored as practical managements to reduce stress-related responses and increase the performance and economic return in caged layers.

We also found that there were no significant differences in the serum levels of the two stress indicators between NS-exposed layers and IMS-exposed layers, although the latter did exhibit the lowest serum ACTH and CORT levels at different sampling points. These results indicated that no extra stress was imposed on layers by IMS exposure. Furthermore, at the end of the experiment, no significant differences in serum ACTH and CORT levels were observed between layers kept at MD (which is a commonly accepted stocking density in China) and LD, suggesting that current conventional stocking density in China (corresponding to MD in our experiment) was within the coping capacities of layers without eliciting deleterious stress in Xiandao green-shell layers.

### Impacts of different stocking densities and sounds on behaviors

Our results demonstrated that LD and IMS were associated with increased preening behavior and decreased feather pecking behavior.

Preening is a self-grooming and feather maintenance behavior. Chickens spend approximately 13% of their daily time on preening (*Dawkins, 1989*). Preening has been interpreted as a comfort-related behavior in hens (*Mollenhorst et al., 2005*; *Iyasere et al.,*

*2022*; *Onbaşılar, 2022*) and the increased preening behavior was associated with reduced negative effects of heat stress in broilers (*Mohammed et al., 2018*, *2022*). However, there are researchers that hold reservations as to whether increased incidence of preening behavior indicates a more comfortable status, or more tendencies to seek comfort (*O'Connor et al., 2011*), or a sign of stereotype associated with stress (*Elston et al., 2000*). In our study significant main effects of sound on preening behavior (frequency and duration) were observed on D12 and D24 with IMS-exposed layers showing the most frequent and longer-lasting preening behavior. Similarly, in a recent research, music stimulus on 10-week old Roman white pullets (10 h per day for 7 consecutive days) significantly stimulated more preening behavior and less feather pecking behavior, suggesting a less stressed status (*Zhao et al., 2023*). Based on our results of serum ACTH and CORT levels in IMS-exposed layers, we concluded that the increased preening activity due to IMS treatment was indicative of more comfortable and less-stressed status in layers.

Feather pecking behavior is a serious animal welfare issue and a main risk factor for feather damage in hens. Feather pecking has been reported to be affected by stocking density (*Zepp et al., 2018*) and high level of noise (*Drake, Donnelly & Dawkins, 2010*; *Gilani, Knowles & Nicol, 2013*). In this study, we found HD and MRS were associated with more frequent feather pecking behavior, which was consistent with the concomitantly elevated serum ACTH and CORT levels. The observed positive relationship between the feather pecking frequency and stocking density are in line with prior research in turkey hens (*Jhetam et al., 2022*) and White Leghorn pullets (*Hofmann et al., 2021*). There is little research on the influence of music on the feather pecking behavior of poultry. *Cabaral, Untalan & Rieta (2017)* implied that the different type of music did not trigger feather pecking behavior among Japanese quails. *Zhao et al. (2023)* find 10-week old Roman white pullets exposed to classic music have less feather pecking behaviors than those in the control group. These findings are consistent with the positive impacts that we observed in the present study. As for the beneficial effects of LD and IMS on reducing feather pecking behavior, we assumed that their influences were additive because the lowest $Freq_{pecking}$ and $Dura_{pecking}$ on D24 were observed in layers subjected to both LD and IMS (group IML).

Altogether, our results showed that LD and IMS were potentially associated with behavior benefits in layers, which may be explored to improve the welfare in Xiandao green-shell layers.

## Impacts of different stocking densities and sounds on cecal bacterial communities

Consistent with previous reports in chickens, our results in Xiaodao green-shell layers confirmed that the microbiota in cecal contents were dominated by Firmicutes, Bacteroidetes, Actinobacteria, Tenericutes and Proteobacteria, in Xiaodao green-shell layers (*Hu et al., 2020*). Different stocking density and sound exposure induced a taxonomic shift in the cecal bacterial communities. Independent of stocking densities, the cecal bacterial communities in IMS-exposed layers were much less diverse than that in NS- and MRS- exposed layers. As for the beta-diversity, cecal bacterial communities tended to separate in response to different sound sources rather than to different stocking densities.

Our results showed that music exposure shifted the two dominant phyla towards an enrichment of Firmicutes with concomitant reduction of Bacteroidetes. Previous reports demonstrated that the cecal microbiota of adult hens (Lohmann Brown Light chickens) consisted of mainly representatives from Firmicutes and Bacteroidetes with approximately equal numbers of both phyla (*Videnska et al., 2014*). In the present study, the Firmicutes to Bacteroidetes ratio (F/B) was around 1.00 for most groups except groups under MRS exposure (0.66) or kept at HD (0.86). The significantly lowered F/B ratio was also observed in the fecal contents of heat-stressed layers (*Zhu et al., 2018*). However, *Wang et al. (2021)* reported a HD-induced significant increase in F/B ratio in layers. We proposed that the inconsistency among results from different studies may be related to differences in chicken species, stress origin, physiological stage of chickens, diet regimens, environment, and sequencing methodology.

Further analysis revealed that *Lactobacillus* from the phylum Firmicutes was the favored genus in the cecal contents of IMS-exposed layers. *Lactobacillus* spp. reduced the intestinal pH level, inhibited pathogens' growth through the production of acetate, lactate, fumarate, hydrogen peroxide, and bacteriocins, and exerted beneficial effects on the host (*Muir, Bryden & Husband, 2000*). Studies in laying hens demonstrated that the significantly increased presence of *Lactobacillus* spp. in the cecum and ileum due to the administration of probiotics (*L. acidophilus*) was closely associated with reduced potential harmful bacteria such as *E. coli*, *clostridia* and *staphylococci* (*Forte et al., 2016*). Significantly increased numbers of *Lactobacilli* and *Bifidobacteria* in the ileum and cecum of chicks due to the treatment of prebiotics and probiotics were associated with decreased numbers of *E. coli*, indicating a balanced intestinal microbiota (*Li, Zhao & Wang, 2009*). Significant enrichment of these two probiotic bacteria in the jejunum were associated with chickens having high body weight and increased anti-inflammatory cytokine levels, indicating improved growth performance and mitigated intestinal inflammation (*Zhang et al., 2022*). In broilers, the increased abundance of *Lactobacilli* and *Bifidobacteria* in the ceca was associated with significantly enhanced weight gain (*Li et al., 2017*). In other animals, *Bravo et al. (2011)* found *L. rhamnosus* (genus *Lactobacilli*) reduced stress-induced CORT and anxiety- and depression-related behavior of the mice, and *Liang et al.*'s *(2015)* results also showed *L. helveticus NS8* (genus *Lactobacilli*) led to lower plasma ACTH and CORT levels, improved chronic restraint stress-induced behavioral and cognitive dysfunction of the rats. In our study, we find layers exposed to IMS did not increase the level of stress hormones, but significantly increased preening behavior and decreased feather pecking behavior. It is speculated that the results may be related to the enrichment of *Lactobacillus* in the intestine of IMS treatment. The significantly enriched *Lactobacillus* in the ceca of IMS-exposed Xiandao green-shell layers may indicate a healthier cecal environment.

Genus *Ruminococcus torques group* was another bacterial biomarker in response to IMS exposure. In a previous study, butyrate-producing *Ruminococcus torques*, belonging to the family Lachnospiraceae, was significantly enriched in the ceca of broilers accompanied by significantly increased concentration of cecal SCFAs (mainly acetate, propionate, butyrate, and lactate) after a supplementation of a prebiotic short-chain arabinoxylans (*Yacoubi et al., 2018*). Members of Lachnospiraceae and Ruminococcaceae were known to be

capable of decomposing recalcitrant substrates (cellulose, hemicellulose, and starch) that were otherwise indigestible by the host (*Biddle et al., 2013*) and in this way converted these compounds into SCFAs which were then absorbed and used as energy by the host. Additionally, results from an *in vitro* study showed that a relatively acidic environment (pH 5.5) strongly favored butyrogenic fermentation accompanied by boosted populations of butyrate-producing bacteria and curtailed growth of *Bacteroides* spp. (*Walker et al., 2005*), which was consistent with the significantly increased F/B ratio observed in IMS-exposed layers. Hence, the IMS-induced significant enrichment of bacteria in genera *Lactobacillus* and *Ruminococcus torques group* helped maintain a more acidic environment in the ceca, improve the host' capacity in degrading complex plant materials, boost the cecal health with a potential to enhance layers' performance. These results coincided with the predicted molecular functions of the cecal bacteria in IMS-exposed layers, which were characterized by significantly enhanced various metabolism pathways and indicated an active metabolic milieu. Collectively, these taxonomic switches in response to music supported our assumption that music, as an auditory enrichment factor, may play an important role in shaping a better cecal bacterial community structure in laying hens.

As for stocking densities, they had impacts, although to a less degree than IMS, on the diversity and community structure of cecal bacterial populations in Xiandao green-shell layers. Two genera from Firmicutes, *Megamonas* (LDA = 3.5) and *Faecalibacterium* (LDA = 4.0), were identified as biomarkers of the cecal bacterial communities in layers kept at LD. *Megamonas*, present mainly in the cecal microbiota of chickens, was likely one of the major propionate producers in Firmicutes (*Sergeant et al., 2014*). *Faecalibacterium* was a well-known butyrate producer and favored the production of butyrate over propionate in chicken cecal contents (*Rowland et al., 2018*). Significantly increased cecal acetic and butyric acids concentrations in chickens were associated with increased antimicrobial capacities against enteric pathogens (*Aljumaah et al., 2020*). The enriched populations of butyrate- and propionate-producing bacteria could improve the cecal heath.

Our results showed that the relative abundance of *Lactobacillus* in the cecal bacterial communities displayed a decreasing trend in response to increasing stocking density. Similarly, a significantly lowered *Lactobacillus* count in the intestinal contents was reported in 42-day broilers kept at high density (*Cengiz et al., 2015*). HD-related alterations in cecal bacterial composition were reflected by significantly enriched bacteria from two genera, *Ruminococaceae UCG 005* and *Ruminococaceae UCG 010*. Limited data obtained in rats demonstrated that the former was a harmful bacterium closely related to oxidative stress (*Qin et al., 2022*), while no detailed function information were available for these two bacteria in chickens.

Altogether, results from the current study highlighted the possibility of improving intestinal microbiota through the strategies of music exposure and density manipulation in Xiaodao green-shell laying hens.

## CONCLUSIONS

The present study confirmed that high density and noise were unfavorable factors associated with the elevated level of serum stress indicators (ACTH and CORT), while low

density and music were two favorable factors capable of stimulating preening behavior and reducing feather pecking behavior in Xiandao green-shell layers. Music- and low-density-mediated alterations in cecal microbiome and functional pathways were characterized by significantly increased bacteria from Firmicutes phylum and *Lactobacillus* genus. Our results provide a preliminary reference for designing management practices through the modulation of stocking density and music exposure to reduce physiological stress, decrease abnormal behaviors, and improve intestinal health in Xiandao green-shell layers. Collectively, the feasibility of reducing stocking density in farms is relatively low in practice, music exposure provides us a cheap and readily available measure to achieve the goal of improving animal welfare by reducing stress and promoting intestinal health by increasing the proportion of beneficial flora.

## ACKNOWLEDGEMENTS

We gratefully acknowledge the support of Heheng Xiandao Poultry Co., Ltd. (Jiangyan District, Taizhou City, Jiangsu Province, China) for providing the experimental site and poultry feed, and the members of Department of Animal Behavior, College of Bioscience and Biotechnology, Yangzhou University (Jiangsu Province, China) for their help.

### Funding

This project was financially supported by the National Natural Science Foundation of China (NSFC) (Grant/Award Number: 31770422). The funders had no role in study design, data collection and analysis, decision to publish, or preparation of the manuscript.

### Grant Disclosures

The following grant information was disclosed by the authors:
National Natural Science Foundation of China (NSFC): 31770422.

### Competing Interests

The authors declare that they have no competing interests.

### Author Contributions

- Shiwen Cao performed the experiments, prepared figures and/or tables, authored or reviewed drafts of the article, and approved the final draft.
- Manhong Ye analyzed the data, prepared figures and/or tables, authored or reviewed drafts of the article, and approved the final draft.
- Wanhong Wei conceived and designed the experiments, authored or reviewed drafts of the article, and approved the final draft.
- Fengping Yang conceived and designed the experiments, authored or reviewed drafts of the article, and approved the final draft.

## Animal Ethics

The following information was supplied relating to ethical approvals (*i.e.*, approving body and any reference numbers):

The Animal Care and Use Committee of Yangzhou University (No. 202104501), Jiangsu Province, PR China provided full approval for this research.

## Field Study Permissions

The following information was supplied relating to field study approvals (*i.e.*, approving body and any reference numbers):

Field experiments were approved by the Animal Care and Use Committee of Yangzhou University (No. 202104501). This study was carried out on a commercial farm, Heheng Xiandao Poultry Co., Ltd. (Jiangyan District, Taizhou City, Jiangsu Province, China) (License number: GB14925-2010).

## Data Availability

The raw data used for analysis is available in the Supplemental Files and at NCBI: PRJNA896261.

## Supplemental Information

Supplemental information for this article can be found online at http://dx.doi.org/10.7717/peerj.18544#supplemental-information.

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
