# Peer review of "Different sound exposures causes alterations in stress-related serum indicators, behaviors, and cecal microbiota of green-shell egg-laying chickens under different stocking densities"

_PeerJ, doi:10.7717/peerj.18544_

## Round 0.1 · original submission · Major Revisions

We have concluded the evaluation of your manuscript with the assistance of three expert reviewers. All have made important suggestions to improve your manuscript as you can see below. I agree with all of these suggestions and therefore ask you to modify your manuscript accordingly. If you do not agree with any of the suggestions please provide a clear justification. All responses should be listed with the appropriate line numbers indicated so that it is easy to see where you have modified the manuscript.

Also, my suggestion would be to indicate in the title that your subject is egg-laying chickens as this was not clear to me at first.

Reviewer 1 ·

Basic reporting

This manuscript by Shiwen Cao et al. employs animal behavior tracking, stress hormone measurement, and microbiota profiling to investigate the impact of sound exposure and stocking density on the performance and welfare of layers. The experimental design is generally clear and well-presented. However, I have a suggestion that could potentially enhance the study:
Major:
1. I find it confusing that the author presents the mean number and SEM in Tables 1 through 4. It would be clearer to present the data using bar graphs that include individual data points.
2. Please clarify how animal behavior is tracked and provide representative images of specific behaviors alongside their quantification.
3. In lines 288-295, the method description should be moved to the appropriate section and include a more detailed biological rationale for microbiota profiling.
Minor
1. Lines 256 -257 are repeating information.

Experimental design

no comment

Validity of the findings

no comment

Additional comments

no comment

·

Basic reporting

In this manuscript, the authors evaluate the effects of stocking density and music exposure on stress-related serum indicators, behavior, and cecal microbiota in chickens. Overall, this paper is well constructed and well written.

Experimental design

The study employed 30-week-old laying hens, which are presumed to be at their optimal level of egg production. This model is more suitable for evaluating the impact of noise and stocking density on egg-laying performance, a crucial aspect that will greatly enhance the value of this study as a reference point for poultry production. It is unclear why the authors did not assess the egg-laying performance. In the event that pertinent records are available, it would be beneficial to include the findings.
It is well documented that both noise and stocking density cause significant stress to chickens. In addition to preening and feather pecking, the stress response has a significant effect on drinking, feeding, stereotypic, lying and standing behaviors. In particular, drinking and stereotypic behaviors are important indicators of stress and stressful environments. It is therefore strongly recommended that the authors obtain these indicators from the video and include them in this study.
Lines 168-170: Why were behavioral recordings and observations made only at 9 a.m.? Chicken preening occurs throughout the day with similar frequency, but feather pecking occurs more predominantly in the afternoon. Furthermore, the limited observation window of 15 minutes could potentially lead to significant sampling errors.

Validity of the findings

No comments.

Additional comments

Line 237: How might the interaction between sound source and stocking density be tested? It is necessary to determine which model is employed. Please provide a detailed description.
Tables: The data should be presented as mean ± standard error of the mean (SEM).

Reviewer 3 ·

Basic reporting

Minor typographical mistakes need correction

Experimental design

One sound treatment was used in each henhouse, correct. This would create a pen effect between henhouses? How was this overcome? The changes observed maybe due to some other environmental factor of each henhouse rather than the music treatment. Seems like the experiment was conducted only once at each barn.

How were the statistics in Tables 4 and 5 completed, ANOVA? However, the microbiome data is compositional in nature and ANOVA is not normally used for this type of data. A test that is designed for compositional data should be used.

Validity of the findings

The results presented in the study show that both stocking density and sound exposure both clearly alter animal behavior and microbiome. To me, this is still a little hard to believe. For example, for duration of feather pecking, instrumental music had a duration of 2, 5, and 5 minutes, while the no sound and road sounds were 4-5 and 5-10 times higher. These are very big differences. In the discussion, the authors do not sufficiently compare their findings specifically to other studies on music on poultry. When feather pecking is discussed, they only compare to studies with "high level of noise". In the microbiome section of the discussion, I find the paragraph on growth promoting antibiotics (lines 445-452) to not supported by the data presented in this paper. I would recommend removing it. The authors need to more fully evaluate the validity of the results, especially in the context of the experimental design as noted earlier.

Lines 355-366 The PICRUSt results are not shown. Why not? If the authors do not want to show these results in a main figure, at least a supplemental figure is needed. Also it is surprising that so many high level (and core) functions are different between the treatment groups. From the taxonomic side, there are some differences at the phylum level, but there are more functional categories that are different, which is puzzling.

Additional comments

Line 199 The HiSeq X10 has a maximum read length of 2x150. However, the sequences available on NCBI have a length of 301. This would likely be from a MiSeq sequencer. Have they given the sequencer name erroneously?
Line 20 “Lots of studies” – too casual of language
Line 70 “proven” not “proved”
Line 102 Can more details be given as to the “increasing importance [of Xiandao green-shell layers] in China”
Line 198 “rRNA gene” not “rDNA”
The methods used in bioinformatics seem cobbled together from a variety of sources. Qiime1 was used in some steps, with is now nearly 10 years old. Why not use Qiime2? Sequence quality filtering was completed, but the filtering criteria are quite permissive.
It is my preference to not use an entire sentence to introduce a Figure or table, as the authors do in lines 245-246, 258-259, 271-272, 297-298, 324-325,
Line 354 is not a complete sentence
Lines 355-356 “PICRUSt” not “PICRUS”
Line 375 “counteract” not “counteracting”
Line 390 I think its important to specify that the Medium density or conventional density is conventional to China and maybe not other countries.
Line 401 “researchers that”
Line 403 correct spelling of “stereotypy”
428 “Actinobacteria”
Line 427 “chyme” is the fluid passed from stomach to small intestine. So that word is not appropriate here.

---

## Round 0.2 · Major Revisions

I have received comments on your manuscript from two reviewers. One of the reviewers is satisfied with the changes that have been made to the document, but the other reviewer has pointed out some serious issues. One of these is the experimental design. I have read through the methods and I am a bit confused as to the experimental design. I think a figure would help perhaps. Another issue that was brought up was the lack of baseline data prior to the start of the experiment to ensure that all chickens were similar. A third issue that was brought up was that the statistical analysis not being appropriate. Please ensure that you respond to each of the concerns raised here and any other issues mentioned by the reviewer. In your rebuttal letter indicate clearly what modifications are made and where they can be found.

·

Basic reporting

no comment.

Experimental design

no comment.

Validity of the findings

no comment.

Additional comments

The authors have addressed all questions I raised. I think it should be accepted as is.

Reviewer 3 ·

Basic reporting

minor corrections need to be made.

Experimental design

Each sound treatment was conducted in one hen house. There were multiple pens in each hen house, but each hen house was separated, so sound treatment is confounded with hen house. Thus we don't know if sound treatment or hen house is the important factor. Also no initial behavior, physiology or microbiome samples were taken, so there is no way to know if the communities in each hen house started in the same condition before the treatment began or not. These are major flaws in the experimental design and would require the experiment to be repeated to increase experimental replication.

Validity of the findings

I disagree with the authors that ANOVA can be used in differential abundance analysis. Even if the data is normally distributed (using a transformation as needed) the data are still compositional, meaning, adding up all the values results in the value of 100%. This does not follow the assumptions of ANOVA and why new tests are actively being developed for differential abundance analysis, like ANCOM. Differential abundance analysis was also completed by the authors with LEfSe, which is a more appropriate test. I recommend deleting tables 4 and 5, and the text in lines 274-276. The same information is included in Fig. 1 and 3, so deleting those two tables does not remove anything from the paper. On the other hand, using ANOVA and t-tests makes more sense to compare alpha diversity metrics, since those metrics are not compositional and can be normally distributed.

Regarding PICRUSt results, I find Figures 4A and S3A very difficult to read. There are so many categories that there are many similar colors. Also there are categories that don’t seem appropriate. All categories related to “Human Diseases” and “Organismal Systems” seem like they do not belong to bacteria, so why are they even considered? And then for panel B, sequence counts were compared. Were sequence counts normalized between samples? I could not find this information in the materials and methods. In all cases of differential abundance, the MRS group is decreased. But its not described this way in the results (line 420).

Additional comments

Lines 108-109: Change to: "Furthermore, due to higher vitamins, trace elements, amino acids, and lower cholesterol and fat, green-shell eggs are considered to be an ideal natural health food (Li et al., 2016).
Line 265: ", good's"
Line 266: "Bray-Curtis"
Line 271: "bacteria"
Line 480-482: Sentence should be in past tense.

---

## Round 0.3 · Minor Revisions

One of the reviewers has some additional observations that need to be addressed. Please ensure you clearly indicate where these have been attended to in the revised version of the manuscript.

Reviewer 1 ·

Basic reporting

no comment

Experimental design

no comment

Validity of the findings

no comment

Additional comments

The author has not addressed my previous concerns.

Reviewer 3 ·

Basic reporting

The authors have thoroughly addressed the concerns. While the concerns still exist, they have made the experimental design more clear for the reader to understand.

Experimental design

For the initial 30 weeks prior to the start of the experiment, how many hens were in a cage? How were the hens allocated to the sound treatment groups? If there were more than 1 hen per cage, were hens from one cage divided between treatment groups?

Validity of the findings

Since the ANOVA tables were removed, the corresponding section of the results should also be edited. Figure 1 is only a illustrative figure. No statistical testing is associated with Fig. 1, so all references to statistical significance should be removed from lines 328-352. The LefSe results could replace this section. Or instead of saying "statistical change" the authors could say a "numerical change". The LefSe portion and the relative abundance portion should be mentioned sequentially and not interrupted by the diversity part (either move LefSe up or the Relative abundance part down)

Additional comments

Lines 161-164: change to: There were 18 cages in each henhouse, totaling 126 layers per room, and the 18 cages were arranged in 6 rows randomly in each henhouse (Fig. S1).

---

## Round 0.4 · accepted · Accept

I am satisfied with the changes that have been made to the manuscript.